# Quantitative determinants of aerobic glycolysis identify flux through the enzyme GAPDH as a limiting step

Alexander A Shestov[1†], Xiaojing Liu[1†], Zheng Ser[1], Ahmad A Cluntun[2], Yin P Hung[3], Lei Huang[4], Dongsung Kim[2], Anne Le[5], Gary Yellen[3], John G Albeck[6‡], Jason W Locasale[1,2,4*]

[1]Division of Nutritional Sciences, Cornell University, Ithaca, United States; [2]Field of Biochemistry and Molecular Cell Biology, Department of Molecular Biology and Genetics, Cornell University, Ithaca, United States; [3]Department of Neurobiology, Harvard Medical School, Boston, United States; [4]Field of Computational Biology, Department of Biological Statistics and Computational Biology, Cornell University, Ithaca, United States; [5]Department of Pathology and Oncology, Johns Hopkins University School of Medicine, Baltimore, United States; [6]Department of Cell Biology, Harvard Medical School, Boston, United States

**Abstract** Aerobic glycolysis or the Warburg Effect (WE) is characterized by the increased metabolism of glucose to lactate. It remains unknown what quantitative changes to the activity of metabolism are necessary and sufficient for this phenotype. We developed a computational model of glycolysis and an integrated analysis using metabolic control analysis (MCA), metabolomics data, and statistical simulations. We identified and confirmed a novel mode of regulation specific to aerobic glycolysis where flux through GAPDH, the enzyme separating lower and upper glycolysis, is the rate-limiting step in the pathway and the levels of fructose (1,6) bisphosphate (FBP), are predictive of the rate and control points in glycolysis. Strikingly, negative flux control was found and confirmed for several steps thought to be rate-limiting in glycolysis. Together, these findings enumerate the biochemical determinants of the WE and suggest strategies for identifying the contexts in which agents that target glycolysis might be most effective.

*For correspondence: locasale@cornell.edu

†These authors contributed equally to this work

Present address: ‡Department of Molecular Cell Biology, University of California, Davis, Davis, United States

## Introduction

Proliferating cells increase their glucose consumption and secrete lactate as opposed to completely oxidizing the glucose in the mitochondria (*Warburg et al., 1927*) and is known as aerobic glycolysis or the Warburg Effect. Currently, this altered metabolism is exploited for diagnostics and is subjected to multiple drug development efforts (*Koppenol et al., 2011*; *Vander Heiden, 2011*; *Hamanaka and Chandel, 2012*). Numerous studies have identified genes such as *KRAS*, *PIK3CA*, and *cMYC* and microenvironments such as hypoxia and hypoglycemia that promote aerobic glycolysis but a complete understanding of the necessary and sufficient biochemical alterations associated with this phenotype is unknown. Furthermore successful translation for biomedical applications is limited by understanding the contexts in which therapies that target glycolysis might be effective.

Computational modeling has a successful history in the study of metabolism (*Rapoport et al., 1976*; *Fell, 1992*; *Schilling et al., 1999*; *Cascante et al., 2002*). Genome-scale stoichiometric models of metabolism have been developed to study the effects of drug targets in human metabolism and have had success in predicting the WE (*Molenaar et al., 2009*; *Vazquez et al., 2010*; *Folger et al., 2011*;

**eLife digest** Cells generate energy from a sugar called glucose via a process called glycolysis. This process involves many enzymes that catalyze 10 different chemical reactions, and it essentially converts glucose step-by-step into a simpler chemical called pyruvate.

Pyruvate is then normally transported into structures within the cell called mitochondria, where it is further broken down using oxygen to release more energy. However, in cells that are rapidly dividing, pyruvate is converted into another chemical called lactate—which releases energy more quickly, but releases less energy overall. Cancer cells often convert most of their glucose into lactate, rather than breaking down pyruvate in their mitochondria: an observation known as the 'Warburg effect'. And while many factors affect how a cell releases energy from pyruvate, it remains unclear what regulates which of these biochemical processes is most common in a living cell.

In this study, Shestov et al. have developed a computational model for the process of glycolysis and used this to investigate the causes of the Warburg Effect. The model was based on the known characteristics of the enzymes and chemical reactions involved at each step. It predicted that the activity of the enzyme called GAPDH, which carries out the sixth step in glycolysis, in many cases affects how much lactate is produced. This suggests that this enzyme represents a bottleneck in the pathway.

Next, Shestov et al. performed experiments where they used drugs to block different stages of the glycolysis pathway, and confirmed that the GAPDH enzyme is important for regulating this pathway in living cancer cells too. In these treated cells, the levels of a chemical called fructose-1,6-biphosphate (which is made in a step in the pathway between glucose and pyruvate) were either very high or very low. Shestov et al. proposed that the flow of chemicals through the glycolysis pathway is controlled by the GAPDH enzyme when the chemicals used by the enzymes upstream of GAPDH in the pathway (which includes fructose-1,6-biphosphate) are plentiful. However, if these chemicals are limited, other enzymes that are involved in earlier steps of the pathway regulate the process instead.

The findings of Shestov et al. reveal that the regulation of glycolysis is more complex than previously thought, and is also very different when cells are undergoing the Warburg Effect. In the future, these findings might help to identify the types of cancer that could be effectively treated using drugs that target the glycolysis process, which are currently being tested in pre-clinical studies.

*Shlomi et al., 2011*). However, a comprehensive quantitative understanding of the WE requires knowledge of enzyme activities and metabolic control.

Therefore, we collected and integrated multiple forms of data into a modeling framework involving flux balances of glycolysis, detailed chemical kinetics based on reaction mechanisms and parameters measured, physico-chemical constraints from thermodynamics and mass conservation, metabolic control analysis, and Monte Carlo sampling of parameter space. We next use mass spectrometry and isotope tracing to probe concentrations and fluxes through the pathway and their responses to several perturbations. Together, we elaborate the determinants of aerobic glycolysis and identify and confirm novel points of regulation in glycolysis that have remained unidentified for over 50 years since the discovery of the pathway.

## Results

### Biochemical kinetic model of aerobic glycolysis

We investigated the kinetics of the glycolytic pathway from glucose uptake to oxidation of pyruvate in the mitochondria or export of lactate out of the cytosol. We modeled each step of the pathway according to enzymatic mechanism and known modes of allosteric control resulting in a set of differential equations (*Figure 1A*, 'Materials and methods', *Supplementary file 1*). While it is not possible to model every possible interaction explicitly, the aim is to capture enough of the pathway so that a large range of experimentally realized measurements can be obtained and relationships between variables can be observed.

Since glycolysis is the most extensively studied biochemical pathway, there is a wealth of information on the kinetic parameters and enzyme expression that govern the equations. Nevertheless, it is also

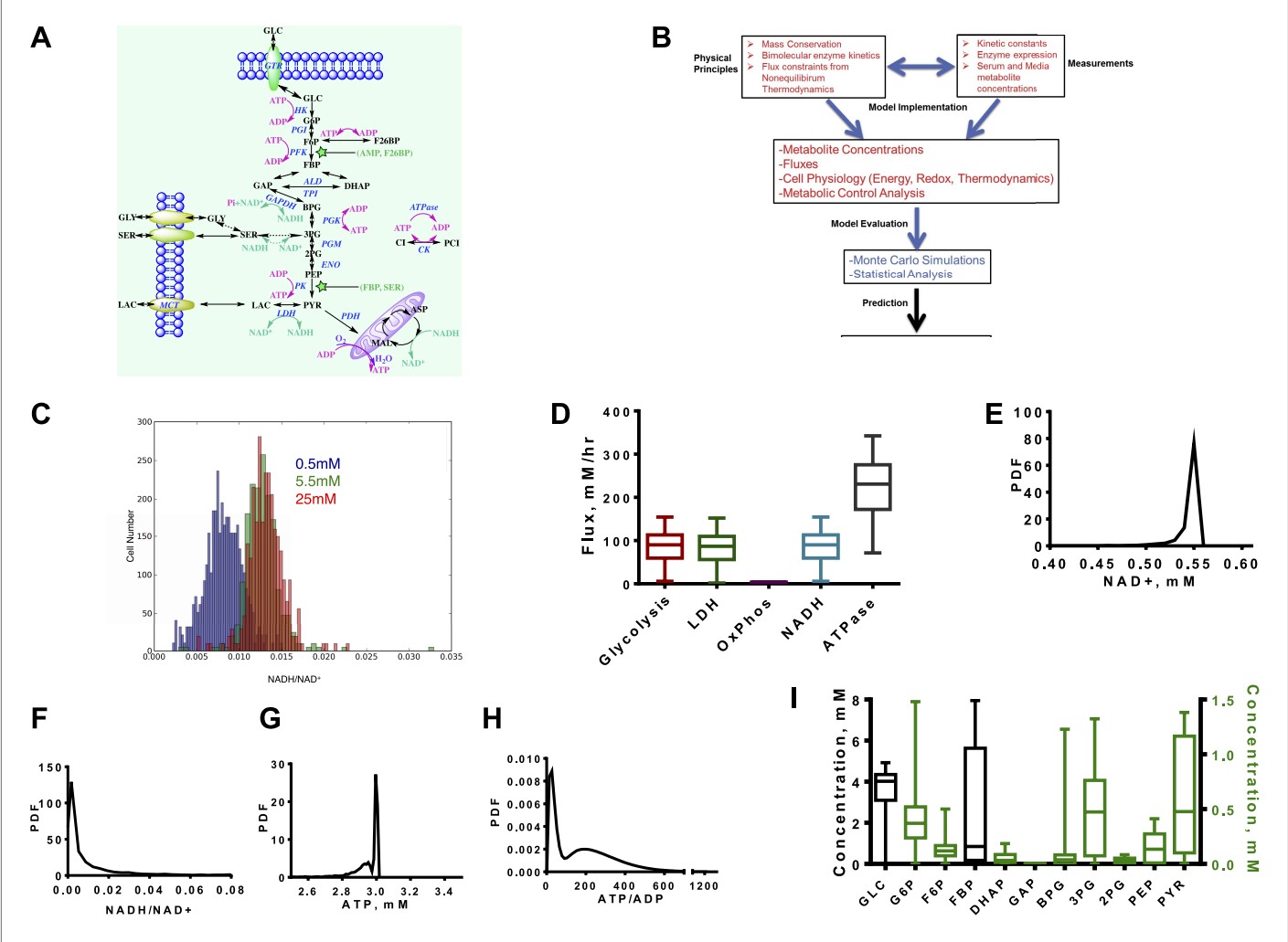

**Figure 1**. A quantitative model and statistical simulation method captures the diversity of metabolic states observed in tumor and proliferating cells. (**A**) Schematic of the glycolysis model with chemical reactions and allosteric points of regulation described. Abbreviations: GLC—glucose, G6P—glucose-6-phosphate, F6P—fructose-6-phosphate, FBP—fructose-1,6,-bisphosphate, F26BP—fructose-2,6,-bisphosphate, GAP—glcyeraldehyde-3-phosphate, DHAP—dihydroxyacetone phosphate, BPG—1,3 bisphosphoglycerate, 3PG—3-phosphoglycerate, 2PG—2-phosphoglycerate, PEP—phosphoenolpyruvate, PYR—pyruvate, SER—Serine, GLY—glycine, Lac—lactate, MAL—malate, ASP—aspartate, Pi—inorganic phosphate, CI—creatine, PCI—phosphophocreatine, GTR—glucose transporter, HK—hexokinase, PGI—phosphoglucoisomerase, PFK—phosphofructokinase, ALD—aldolase, TPI—triosephosphoisomerase, GAPDH—glyceraldehyde-phosphate dehydrogenase, PGK—phosphoglycerate kinase, PGM—phosphoglycerate mutase, ENO—enolase, PK—pyruvate kinase, LDH—lactate dehydrogenase, MCT—monocarboxylate transporter, PDH—pyruvate dehydrogenase, CK—creatine kinase. (**B**) Overview of the algorithm and simulation method. (**C**) Measured values of the NADH/NAD+ ratio across a population of MCF10A breast epithelial cells. Three values of glucose concentration are considered (0.5 mM blue, 5.5 mM green, and 25 mM red). (**D**) Calculated fluxes (mM/hr) for glycolysis rate (Glycolysis) are defined as the rate of glucose to pyruvate (per molecule of pyruvate), pyruvate to lactate flux (LDH), rate of oxygen consumption (OxPhos), rate of NADH turnover (NADH), and ATP turnover (ATPase). (**E**) Calculated probability density function (PDF) of NAD+ concentrations. (**F**) Calculated probability density function (PDF) of NADH/NAD+ ratio. (**G**) Calculated probability density function (PDF) of ATP concentrations. (**H**) Calculated probability density function (PDF) of ATP/ADP ratio. (**I**) Box plots showing the distribution of concentrations computed from the simulation for each intermediate in glycolysis.

not possible to capture cellular physiology in any biochemical model with single values of kinetic parameters (***Daniels et al., 2008***). This difficulty arises from the tremendous amount of heterogeneity within cells at multiple levels. The origins of this heterogeneity vary from genetic variation observed across cancer types, tumor types, differences in signaling mechanisms that affect post-translational modifications in each cell, and the differences in microenvironmental pressures (e.g., the oxygen availability) that each cell within a given tumor experiences, as well as the inherent cell to cell variation common to all cells (***Marusyk et al., 2012***). Therefore, we developed an integrated algorithm to evaluate

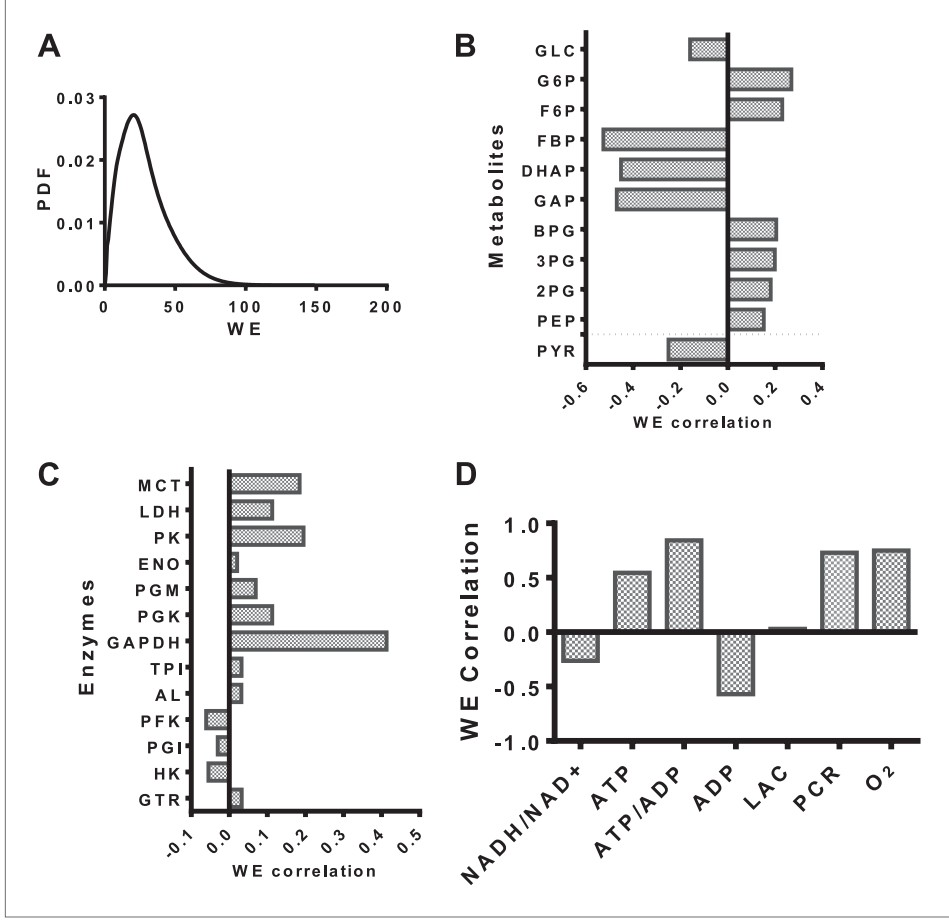

**Figure 2**. Evaluation of the statistics of the Warburg Effect and relationships to other variables in metabolism. (**A**) Probability density function (PDF) of the Warburg Effect (WE) defined as the ratio of flux through LDH to that of flux into the mitochondria. (**B**) Pearson correlations of intermediate metabolite levels in glycolysis with the extent of the Warburg Effect (WE). (**C**) Pearson correlations of the expression levels of glycolytic enzymes with the extent of the Warburg Effect (WE). (**D**) Pearson correlations of coupled metabolic parameters with the extent of the Warburg Effect (WE).

the statistics of the kinetics of glycolysis that accounts for the possible variation within metabolism (*Figure 1B*, 'Materials and methods').

First the model is constrained using mass conservation constraints that conserve the balance of glucose, redox state, and energy status. Next, thermodynamic constraints are used to constrain fluxes according to the free energy of the reactions determined by Haldane relationships. These physical constraints are combined with the kinetic mechanisms that define each step of glycolysis and the chemical reactions involving the redox-associated metabolites $NAD^+$, and NADH and energy-associated metabolites ATP, ADP, and AMP. Next, the model is constrained to expression data so that protein concentrations are subsumed in the Vmax values and chosen from typical concentrations in cancer cells, measured kinetic parameters, and measured concentrations of nutrients such as glucose, oxygen, and total intracellular adeno-nucleotide concentration.

At this stage, the model is subjected to a thorough statistical analysis. A Monte Carlo simulation is conducted for which the parameters within the model are randomized and resulting differential equations are solved. The distributions for each parameter are chosen to capture observed ranges of variation. After each simulation, numerical stability and thermodynamics are assessed and all simulations that are unstable or appear thermodynamically infeasible (i.e., a positive net flux through glycolysis is required) are rejected. In each simulation, concentrations, fluxes, metabolic control coefficients, and thermodynamic quantities are computed and recorded. This statistical analysis explores

the space of glycolysis in the context of the Warburg Effect. By assessing the statistics, inferences can be made on the determinants of the Warburg Effects and its context dependence to pharmacological intervention and nutrient environment. For experimental confirmation of the model, we first utilized a recently developed a NADH/NAD⁺ fluorescent reporter (*Hung et al., 2011*) and measured the ratio of NADH/NAD⁺ across a population of cells where some experience hypoxia and others glucose deprivation (*Figure 1C*). This variation in nutrient availability across individual cells occurs in epithelial cell cultures due to differences in diffusion and available cell surface area (*Sheta et al., 2001*). The distribution is plotted for three media concentrations ranging from extreme hypoglycemia to the hyperglycemic conditions typically used in cell culture (0.5 mM, 5.5 mM, and 25 mM).

Next we noted that the distribution of resulting fluxes (*Figure 1D*) was first found consistent with known measurements. An analysis of the simulation revealed a distribution of NAD⁺ levels and NADH/NAD⁺ redox potential consistent with those observed experimentally (*Figure 1E,F*). ATP levels peaked around 3 mM with little variation (*Figure 1G*) and the ATP/ADP energy state is observed to be bimodal (*Figure 1H*). Concentrations along the glycolytic pathway varied with means similar to those measured in normal tissues with fructose-1,6-bisphosphate (FBP) being most variable (*Figure 1I*). Together, these findings indicate that these simulations capture the range of measured cellular concentrations and provide confidence for further assessment of the behavior within the model.

## Evaluation of the Warburg Effect and its relationship to metabolic variables

Having developed a model of glycolysis and its regulation, we assessed the relationship of aerobic glycolysis to other characteristics of glycolysis. The distribution of values (*Figure 2A*) of the Warburg Effect (the ratio of flux to lactate over that entering the mitochondria) ranged from less than one (primarily oxidative metabolism) to over 95% of glucose being converted to lactate. The dynamic range over which the Warburg Effect was observed in our model allowed us to investigate to other variables in metabolism. We correlated the calculated value of the Warburg Effect with metabolite concentrations of intermediates in glycolysis. From an analysis of these correlations, a pattern within glycolysis emerges. The levels of intermediates in the beginning steps in glycolysis positively correlate with the Warburg Effect, the levels leading up to the oxidation of glyceraldehyde-3-phosphate (GAP) by GAPDH negatively correlate with the Warburg Effect, and those following GAPDH positively correlate with the Warburg Effect (*Figure 2B*) suggesting together that a bottleneck exists in the pathway that determines the extent of fermentation. An analysis of the correlation of enzyme expression with the Warburg Effect revealed that the enzyme expression for any single step within glycolysis did not completely correlate with the Warburg Effect. However, across glycolysis, GAPDH, the enzyme that carries out oxidative phosphorylation of glyceraldehyde-3-phosphate to yield NADH and 1,3-diphosphoglycerate most strongly correlated with aerobic glycolysis (*Figure 2C*). Notably many of the correlations although significant, appear not very strong indicating that the expression of individual enzymes is not sufficient for induction of aerobic glycolysis. The model also captures many of the activities that are known to correlate with aerobic glycolysis. These include glucose transport, pyruvate kinase, and lactate dehydrogenase activities. Interestingly, the expression of enzymes such as hexokinase and phosphofructokinase were either uncorrelated or negatively correlated with the extent of aerobic glycolysis likely indicating their role in creating bottlenecks at other points along the pathway. Together, these findings identify enzyme expression patterns that determine the extent of the Warburg Effect.

We next investigated the relationship between the Warburg Effect and other physiological variables including the NADH/NAD⁺ redox status, the energy state defined as the ATP/ADP ratio, lactate, oxygen levels, and phosphocreatine levels (*Figure 2D*). Oxygen concentration, ATP, and phosphocreatine levels positively correlated with the Warburg Effect and NADH/NAD⁺ redox status and NADH levels in the cytosol negatively correlated with the Warburg Effect suggesting that positive and negative feedback inherent to the circuitry of glycolysis contributes to buffering the Warburg Effect. Together, these results identify multiple relationships between the extent of aerobic glycolysis and measurable variables in metabolism.

## Flux control in glycolysis

After assessing how metabolic parameters and concentrations within glycolysis determine the flux to lactate, we next investigated how each node within glycolysis exerts its control on the Warburg

Effect. Metabolic control analysis (MCA) provides a mathematical framework for evaluating the extent that a change in metabolic activity affects a given flux. In MCA, a pure rate-limiting step occurs when the flux control coefficient (FCC) (Supplementary Information) is one at that step and zero at all other steps. In most cases, values of FCCs are distributed with gradual values across a pathway. We carried out a metabolic control analysis using MCA within our statistical algorithm to investigate the influence of each node in glycolysis on lactate flux across the ensemble of statistical realizations of glycolysis (*Figure 3A*). We first computed the distribution of FCCs for lactate production for each step in the pathway (*Figure 3B*). It was found that for each step within glycolysis, the average control exerted was near zero. Steps early in glycolysis involving enzymes Hexokinase (HK), Phosphoglucoisomerase (PGI), and Phosphofructokinase (PFK) exhibit both positive and negative control, and GAPDH on average exhibits the most positive control on the flux through the pathway. In addition to steps within glycolysis, ATPase activity exerts the most influence over lactate production with oxygen consumption having less of an affect. Together, these results both confirm the long-standing hypothesis that ATPase activity is the most prominent rate-determining step in glycolysis but also suggest that GAPDH can often exert a large control on the flux to lactate. In

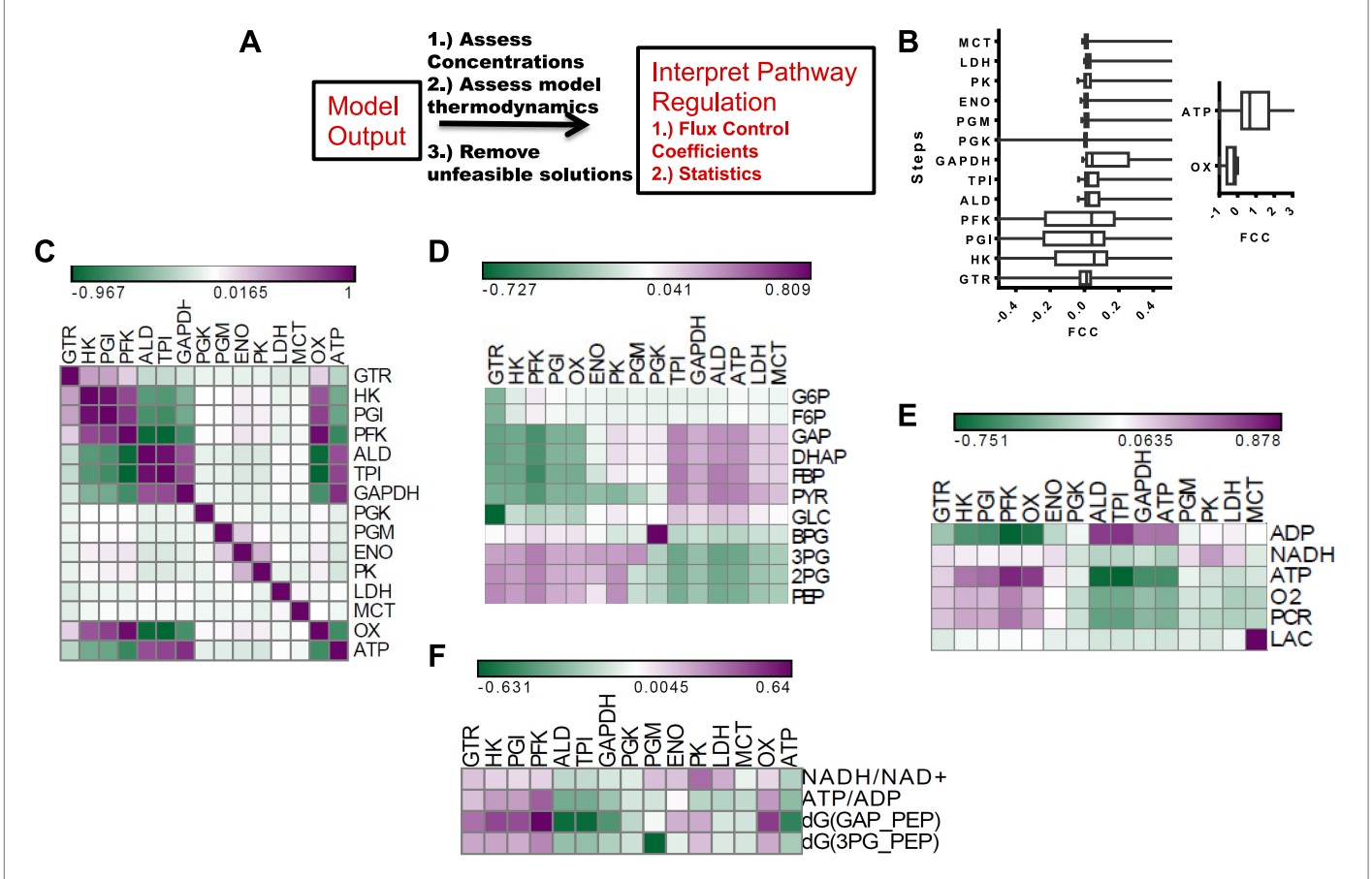

**Figure 3.** Metabolic control analysis and its relationship to metabolic variables. (**A**) Schematic of workflow for global sensitivity analysis. After the model is constructed and feasible solutions obtained, each realization of glycolysis is subjected to metabolic control analysis (MCA). The resulting analysis is then subject to a statistical evaluation. (**B**) (left) Box plots of flux control coefficient (FCC) for lactate production for each enzymatic step in glycolysis (FCC = $dlnJ_{lac}/dln\ E_i$) where $J_{lac}$ is the rate of pyruvate conversion to lactate, and $E_i$ is the $i^{th}$ enzyme in glycolysis for each step of glycolysis. (right) Box plots of flux control coefficient (FCC) for lactate production for Oxygen consumption (OxPhos) and ATP consumption (ATP). (**C**) Pearson correlations between lactate FCC values for each step in glycolysis. Heat map is colored ranging from the minimum value (green) to the maximum value (purple). (**D**) Pearson correlations between metabolite concentrations in glycolysis and lactate FCC values for each step in glycolysis. Heat map is colored ranging from the minimum value (green) to the maximum value (purple). (**E**) Pearson correlations between metabolic parameters and lactate FCC values for each step in glycolysis. Heat map is colored ranging from the minimum value (green) to the maximum value (purple). (**F**) Pearson correlations between ratios and lactate FCC values for each step in glycolysis.

addition, an analysis of the statistics indicates that depending on the context, any point along the pathway can exert large flux control on lactate production with some steps (e.g., pyruvate kinase) less likely to exert control over the Warburg Effect than others. Although the finding that pyruvate kinase typically does not exert substantial control over glycolysis may appear surprising, this finding is consistent with studies that have observed only modest changes in glycolytic flux due to changes in pyruvate kinase activity (*Christofk et al., 2008*; *Israelsen et al., 2013*).

To further investigate the contexts in which steps along glycolysis can be limiting, we correlated the FCCs with one another and carried out a hierarchical clustering that revealed modules of co-occurring flux control (*Figure 3C*). It was found that enzymes residing in proximal regions of upper glycolysis tend to have positively correlated flux control. However, for lower glycolysis the flux control is uncorrelated suggesting that when a step is limiting in lower glycolysis, it can be more readily disrupted. This behavior is in contrast with that of upper glycolysis where limiting steps co-occur. Furthermore, an analysis of ATPase and Oxygen consumption activities reveals their control to be related to the control of enzymes clustered at different points in glycolysis (*Figure 3C*) indicating that flux control exerted by these ancillary fluxes are tied to different regions of glycolysis. Since these simulations yield for each realization of glycolysis, a set of concentrations, fluxes and flux control coefficients, relationships between flux control coefficients and measurable concentrations can be obtained. An analysis of these relationships using hierarchical clustering (*Figure 3D*) revealed a bi-modal relationship in glycolysis where increases in metabolites in upper glycolysis led to steps in lower glycolysis exerting flux control and increases in lower glycolysis resulted in enzymes in upper glycolysis exerting flux control. Further analysis of FCCs and metabolic parameters (*Figure 3E,F*) revealed additional relationships. Together, these findings yield a comprehensive map of the flux control in glycolysis and its connections to metabolic variables.

## Experimental flux control in glycolysis

Surprisingly, in several instances, negative flux control is observed for several enzymes that have been previously thought to be the rate-limiting steps. Negative flux control implies that inhibiting the enzyme would actually increase the rate of flux of glucose to lactate. Given the surprising findings on flux control in glycolysis, we sought to experimentally investigate these predictions. We therefore considered systematic perturbations to glycolysis in cancer cells.

We first devised a combined flux profiling and metabolite profiling method using $^{13}$C isotope tracing from glucose (*Figure 4A*). Cells were incubated with U–$^{13}$C glucose and media were extracted at different time points and subjected to high resolution LC-MS analysis (*Liu et al., 2014*). The linearity of the time course allows for an exact measurement of the flux from glucose to lactate. We next carried out direct measurements of glycolysis state and flux control by perturbing glycolysis acutely with pharmacological agents and subsequently measuring metabolite levels and the flux from glucose to lactate. Genetic manipulations such as RNA interference were not possible since the measurement of dose-dependent acute effects was necessary. In each case, the compounds considered have been reported to exhibit direct inhibition of the enzyme in question and to our knowledge do not directly inhibit other enzymes in glycolysis. Nevertheless, the general specificity of these compounds is not established and off target effects that exist are likely reduced by considering acute treatments.

We considered inhibiting glycolysis at three separate points along the pathway that are predicted to have widely variable flux control of the pathway. In the beginning of glycolysis we used 3PO, a compound that targets PFK2 thus inhibiting the phosphofructokinase step (*Schoors et al., 2014*). Next, we considered iodoacetate (IA), a compound that targets GADPH (*Campbell-Burk et al., 1987*). Finally FX11, a compound that targets LDH was considered (*Granchi et al., 2011*).

In each case (*Figure 4B–D*) differential, complex, nonlinear responses of metabolite levels to inhibiting glycolysis were observed. The exception was treatment with IA (*Figure 4C*) which exhibited an expected accumulation of intermediates upstream of GAPDH and depletion of intermediates downstream of the target. We next measured directly the flux control coefficients for each compound. An analysis of the flux control (*Figure 4E–G*) revealed several interesting responses. Strikingly, as predicted by the model, a negative flux control was observed for targeting PFK implying that inhibiting this step in these circumstances increases glycolytic flux and Warburg Effect (*Figure 4E*). The largest flux control was observed with inhibition of GAPDH (*Figure 4F*). Inhibition of LDH as predicted also resulted in little flux control (*Figure 4G*). Together, these findings confirm the mechanisms of flux

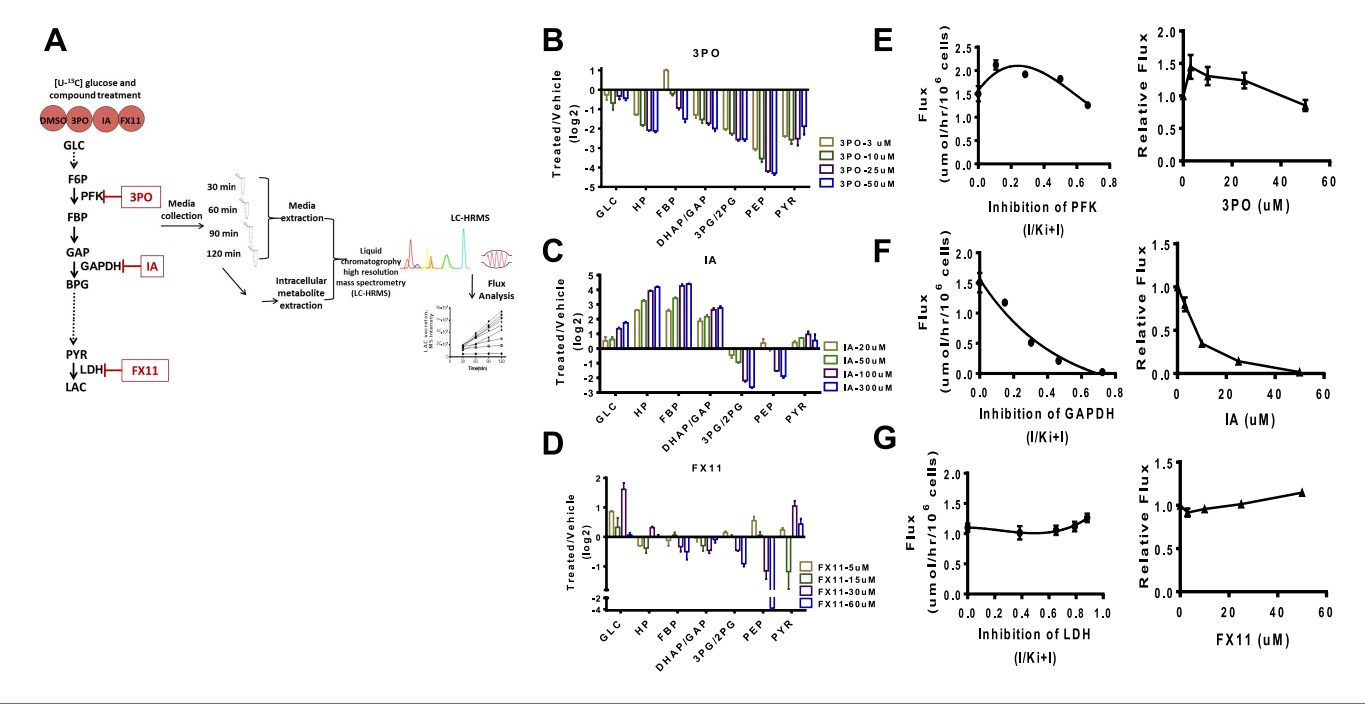

**Figure 4.** Experimental flux control coefficients. (**A**) Schematic of experimental flux control analysis. Cells are pre-incubated with [13]C glucose and treated with differing concentrations of inhibitors that target glycolysis at different points in the pathway. Media and intracellular metabolites are collected, subjected to (liquid chromatography high resolution mass spectrometry) LC-HRMS, and subjected to flux analysis. (**B**) Changes in metabolite levels observed from treatment with 3PO an inhibitor of PFK2. (**C**) Changes in metabolite levels observed from treatment with IA an inhibitor of GAPDH. (**D**) Changes in metabolite levels observed from treatment with FX11 and inhibitor of LDH. For **B**–**D**, the logarithm ($\log_2$) of the fold change of treated to vehicle across intermediates in glycolysis is shown for each concentration of compound denoted in the figure legend. Abbreviations are the same as described in *Figure 1* except that HP denotes all hexose phosphates that were measured and not distinguished in the current mass spectrometry method. (**E**) Lactate flux from glucose as a function PFK2 inhibition. (**F**) Lactate flux from glucose as a function GAPDH inhibition. (**G**) Lactate flux from glucose as a function LDH inhibition. For **E**–**G**, the plot on the left shows the measured glucose to lactate flux as a function of the estimated fraction of enzyme inhibited (left) inhibitor concentration (right).

control of glycolysis predicted by the model and demonstrate the novel regulation of glycolytic flux that can be differentially perturbed by pharmacological compounds.

## FBP status and signatures of glycolysis state

We have thus far observed experimentally the dynamic behavior of flux control in glycolysis including both positive and negative flux control in the pathway and a high variability of flux control depending on the point of inhibition in the pathway. Having established the complex relationships between glycolytic flux and susceptibility to specific targeted inhibition of the pathway, we sought to investigate whether there was any predictive capacity and new mechanisms of biochemical regulation related to these findings.

We noticed that Fructose-(1,6)-bisphosphate (FBP) levels exhibited highly dynamic and counterintuitive behavior with each drug perturbation. An analysis of metabolite levels across a series of 14 conditions in triplicate involving pharmacological perturbations of PFK2, LDH, and GAPDH at different concentrations and two separate vehicle treatments (*Figure 5A*) revealed large magnitude, dynamic responses in FBP levels. We next investigated the extent that FBP levels could characterize the metabolic state of glycolysis. The simulated PDF of FBP levels exhibited a bimodal distribution (*Figure 5B*) consisting of a state of low FBP where the concentration was in the high micromolar range. In addition, there existed a state of FBP where the concentration was several orders of magnitude high in the millimolar range. When demarcating the experiments into two groups (high and low FBP) we correlated the levels with lactate flux experimentally (*Figure 5C,E*) and found the results to match those observed computationally (*Figure 5D,F*). In addition, correlations in these two states with the remainder of the

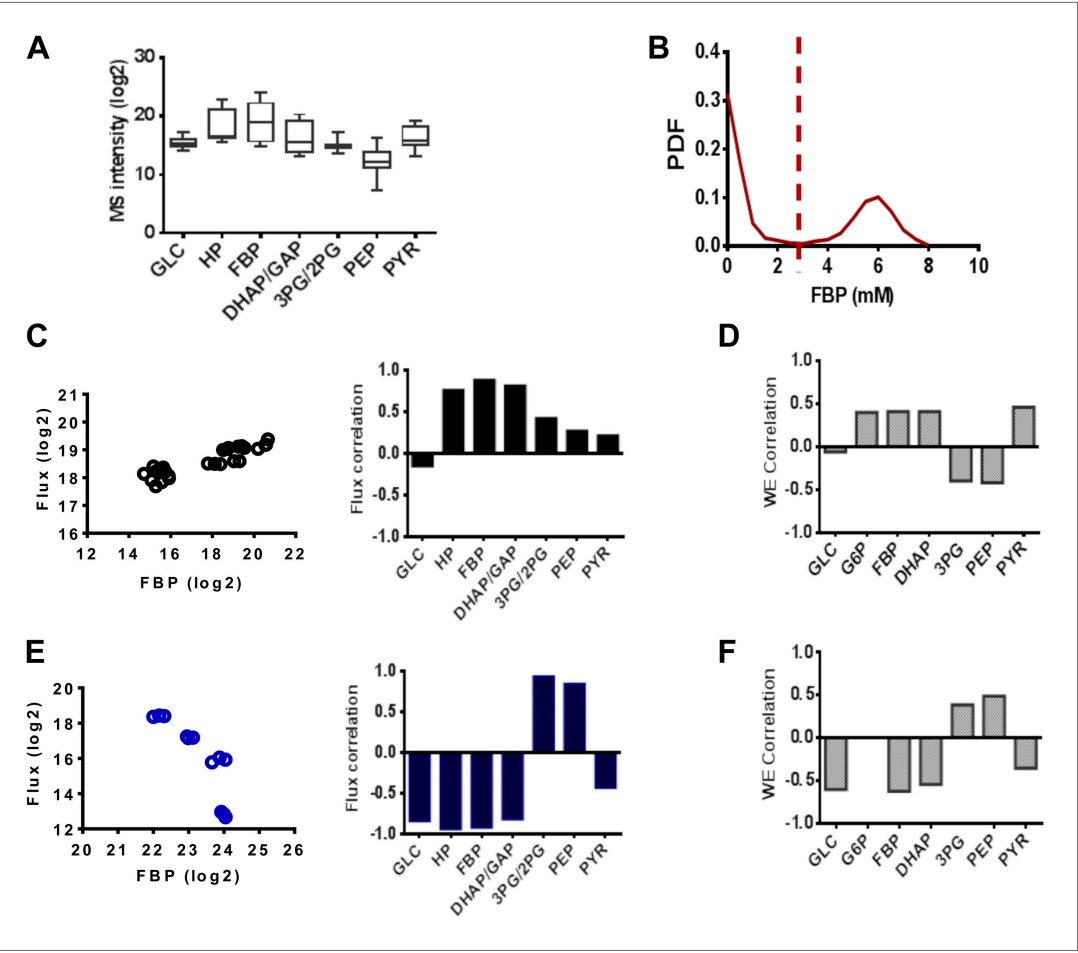

**Figure 5**. FBP levels predict distinct mechanisms in glycolysis. (**A**) Variation of metabolite levels across glycolysis over 14 conditions in triplicate resulting in 42 independent experiments involving cells growing in basal conditions and those with differing extents of inhibition of glycolysis from results in **Figure 4**. (**B**) Simulated distribution of FBP levels in glycolysis. (**C**) Correlation of lactate flux with measured glycolytic intermediates for low FBP levels. The left panel shows data for FBP and right panel reports the values of the Spearman correlation coefficients for each metabolite. (**D**) Simulated correlation of lactate flux with metabolite levels of glycolytic intermediates in conditions of low FBP levels. (**E**) Correlation of lactate flux with measured glycolytic intermediates for high FBP levels. The left panel shows data for FBP and right panel reports the values of the Spearman correlation coefficients for each metabolite. (**F**) Simulated correlation of lactate flux with metabolite levels of glycolytic intermediates in conditions of low FBP levels.

concentrations of glycolytic intermediates agreed well with experiments. Together these findings suggest in addition to the model being able to capture a diverse set of experimental metabolic conditions, the FBP status in cells was able to determine the state of glycolysis and its rate-limiting steps.

## A unified model of aerobic glycolysis

Together, the combined computational model, metabolite profiling, and flux analysis points to a different picture of glycolysis (**Figure 6**). When the levels of FBP are low, metabolite levels of glycolytic intermediates tend to be more evenly distributed across the pathway. In this circumstance, flux through the pathway is controlled largely through the initial steps in glycolysis involving hexokinase and phosphofructokinase. Bottlenecks downstream of these canonical rate-limiting steps are not affecting flux through the pathway. Under these conditions, FBP is also not allosterically activating pyruvate kinase. In contrast, when the levels of FBP are high, there is a disconnect in the relative concentration of glycolytic intermediates that is marked by a separation between upper and lower glycolysis at the GAPDH step. In this case, there is an accumulation of intermediates in upper glycolysis, most notably

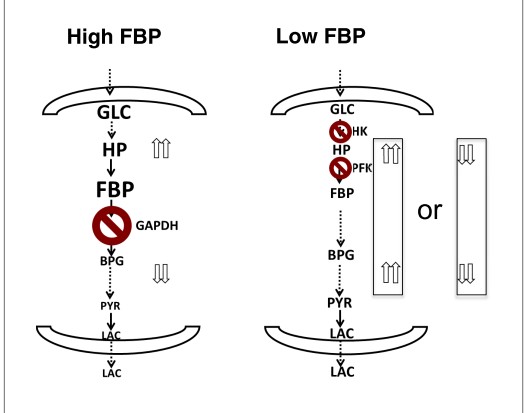

**Figure 6**. A unified model of aerobic glycolysis. A unified picture of flux control in aerobic glycolysis. (left) Under conditions where there is an accumulation of intermediates in upper glycolysis and depletion of intermediates in lower glycolysis a bottleneck exists at the step involving GAPDH. This bottleneck is due to the status of energy and redox metabolism and the thermodynamics of the pathway that together mediate the flux through GAPDH. As a result, inhibiting flux through glycolysis is most sensitive to a perturbation in GAPDH activity. (right) Under conditions where the metabolites in glycolysis are distributed more evenly with levels together being either high or low, no such bottleneck exists. Instead flux through glycolysis leading to lactate production is most determined by the canonical pacemaking steps in glycolysis involving PFK and HK. The relative levels of glycolytic intermediates are denoted by the size of the text.

FBP and a depletion of intermediates downstream of GAPDH. Under these circumstances, GAPDH exerts control as the most rate-determining step in the pathway since increased activity through GAPDH will serve then to create a balance along the pathway by pulling the metabolites from upper glycolysis into lower glycolysis. Under these conditions, the beginning steps of glycolysis can exert negative control on the pathway since inhibiting them will result in even greater increases in the intermediates in upper glycolysis. This state has analogies with a recently observed state of glycolysis observed in yeast where an accumulation of FBP leads to cellular toxicity (*van Heerden et al., 2014*). Notably in this case, in order to balance the fluxes in higher and lower glycolysis, an imbalance in the concentrations of metabolites in upper and lower glycolysis results.

## Discussion

Together, our analysis yields a comprehensive, quantitative framework for understanding glycolysis and its regulation in the context of the Warburg Effect. Historically, glycolysis is thought to have a rate-limiting step at several points in the pathway (*Chance and Hess, 1956*; *Wu, 1965*). These points correspond to positions in the pathway where large free energy differences arise including the ATP-coupled enzymes Hexokinase, Phosphofructokinase, and Pyruvate Kinase (*Rose and Warms, 1966*; *Rapoport et al., 1976*; *Hue and Rider, 1987*). Surprisingly, we identified a strong context-dependence with both positive and negative control in glycolysis at each of these steps. In the case of inhibiting flux through PFK, the observed negative control in certain conditions implies that inhibiting this point in the pathway can lead to increased rates of fermentation. This finding provides a possible explanation for why its efficacy may be more prevalent in stromal cells (*Schoors et al., 2014*).

Unexpectedly, GAPDH was found to be a recurrent rate-controlling step in aerobic glycolysis. This finding, first documented to our knowledge in parasitic bacteria feeding on high glucose (*Bakker et al., 1999*), is in contrast to the longstanding notion that GAPDH is not a rate-determining enzyme in glycolysis with the activity of enzymes such as hexokinase, phosphofructokinase, and pyruvate kinase thought to be more controlling. In the case of GAPDH, the bottleneck occurs due to its unique placement in the pathway where it can be regulated by ATP, NAD$^+$, and the levels of glucose-derived intermediates in the pathway that affects the thermodynamics of glycolysis. There are multiple mechanisms that lead to this finding. ATP consumption has previously been reported to control the rate of glycolysis and this effect likely occurs to some extent through GAPDH (*Racker, 1976*; *Locasale and Cantley, 2011*; *Lunt and Vander Heiden, 2011*). NAD$^+$ regeneration that is mediated by the malate–aspartate shuttle and lactate dehydrogenase also affects flux through glycolysis (*DeBerardinis et al., 2008*; *Locasale and Cantley, 2011*). In addition, the activity of the pathway upstream and downstream of GAPDH changes the balance of the levels of intermediates in glycolysis and results in driving the thermodynamics of the reactions out of equilibrium also can result in greater flux control (*Noor et al., 2014*). Each of these mechanisms separately or together acts to allow for GAPDH to exert flux control over the glycolytic pathway.

Enzymes along glycolysis that are believed to control flux have many documented regulatory mechanisms. For example pyruvate kinase and phosphofructokinase have numerous small molecule effectors and post-translational modifications that affect their activities (*Mor et al., 2011*; *Chaneton and Gottlieb, 2012*;

*Keller et al., 2012*; *Yi, et al., 2012*). It is notable that GAPDH also is subject to multiple forms of regulation including post-translational modifications such as nitrosylation and reactive oxygen species (ROS) that interacts with the catalytic cysteine in GAPDH to inhibit its activity (*Gaupels et al., 2011*; *Tristan et al., 2011*; *Moellering and Cravatt, 2013*). In the context of ROS, it is tempting to speculate that alterations in ROS could lead to selective modulation of glycolytic flux as has been suggested to occur with pyruvate kinase (*Anastasiou et al., 2011*). While ROS-mediated inhibition is unlikely in conditions of exponential growth, it may be more apparent in physiological conditions of hypoxia and glucose deprivation with higher concentrations of ROS. Another critical aspect of the flux control that GAPDH exerts over the glycolytic pathway is high expression of GAPDH in cells undergoing aerobic glycolysis. Indeed, reports of quantitative protein abundance in mammalian cells have identified enzymes in the pathway glycolysis as the most highly expressed collective of proteins in cells (*Moghaddas Gholami et al., 2013*). Interestingly, it was found that within glycolysis, GAPDH is often the mostly highly concentrated protein in glycolysis suggesting that the role of this high expression in these cases is to support the increased amount of glycolytic flux in these cells. Nevertheless, although mechanisms that regulate phosphofructokinase and pyruvate kinase for example have been shown to mediate cell growth and proliferation, whether regulatory mechanisms of GAPDH have functional roles in cell growth and proliferation related to aerobic glycolysis is not known.

Two states of glycolysis were observed with different extents of flux control, tendencies for aerobic glycolysis, and concentration patterns along glycolysis. Notably, the FBP levels have implications on the activity of pyruvate kinase that is also allosterically activated by FBP. At low FBP concentration, pyruvate kinase is not activated by FBP but this occurs only in the high FBP state. This finding could have implications in understanding the contexts in which pharmacologically activating pyruvate kinase may have efficacy. Furthermore, it remains to be seen if any signature of these metabolite states could manifest in the alterations of peripheral metabolism involving pathways whose fluxes emanate from glycolysis. If this were the case, then a tempting possibility would be that these states of glycolysis could be predicted from measurements of peripheral metabolites that could be excreted into circulation allowing for the possibility of developing serum biomarkers for the status of glycolysis in tumors beyond what can be resolved with positron emission tomography using radioactive glucose.

Finally, the surge of interest in metabolism and its contribution to pathogenesis has created an expectation that therapeutics that target glucose metabolism will be clinically successful. Targeting glycolysis in metabolism has raised interest but is limited by the development of biomarkers that could determine the contexts in which targeting glucose metabolism in malignancy would be efficacious (*Vander Heiden, 2011*; *Galluzzi et al., 2013*; *Vander Heiden, 2013*). From our model analysis, the consequences of inhibiting glycolysis appear enormously complex that limit biomarker development due to the nonlinear mechanisms that determine the response of the pathway to a drug perturbation. Nevertheless, with the development of these predictive models that can capture diverse behaviors of glycolysis, it is worth considering whether they may have predictive capacity in pre-clinical and clinical settings.

## Materials and methods

### Kinetic model of glycolysis

The model includes a compartment involving enzymes in glycolysis and additional compartments involving with reactions that are coupled to glycolysis. The following compartments are considered in addition to the enzymes in glycolysis are considered:

1. Glucose uptake through the glucose transporter,
2. Lactate dehydrogenase (LDH) activity and lactate transport through the monocarboxylate transporter (MCT),
3. Oxygen consumption, transport and activity of oxidative phosphorylation (OxPhos),
4. ATP-buffering mechanisms involving adenylate Kinase, creatine Kinase, and ATPase activity,
5. NADH/NAD$^+$-mediated mitochondrial shuttles including the Malate–Aspartate Shuttle and the Glycerol-3-Phosphate Shuttle.

More generally glycogen metabolism, pentose phosphate pathway and alanine biosynthesis could also be included but are omitted since the conclusions drawn in this study do not depend on their activities. A schematic representation of the kinetic model and resulting network is shown in *Figure 2A*.

We first introduce notation and other conventions. Each symbol indicates the respective concentration. A subscript '0' refers to the steady-state value. Reaction rates are expressed in millimoles per hour per unit intracellular volume. Together with initial state vectors $C_0 \in R^N$, $V_o \in R^M$, of metabolites and fluxes respectively.

The dynamics of the system are therefore formulated as initial value problem for ordinary differential equations (ODEs). Starting parameter values based on published data are defined in **Supplementary file 1**. Reaction rates are defined through a consideration of the respective enzyme mechanisms with additional feed forward and feed back regulation giving rise to allosteric activation and inhibition.

Therefore, the model is designed to describe: (a) steady-state and dynamic behavior of energy metabolism, including the Warburg effect ($W = \dfrac{J_{Lac}}{J_{Ox}}$), energy state (ATP/ADP ratio), redox state (NADH/NAD$^+$ ratio) together with glucose, lactate, and oxygen supply through exchange fluxes from the intercellular and extracellular compartment, (b) steady states and time courses of all variables of the metabolic network, and (c) effects of metabolic parameter perturbations on the overall system output. Cellular energy homeostasis through ATP is supported by several mechanisms with different relaxation times and regulation mechanisms: (a) directly by glycolysis that converts glucose into intracellular pyruvate, (b) mitochondrial respiration through consumption of pyruvate and oxygen via the TCA cycle, and (c) the buffering effect of creatine kinase that facilitates the reversible reaction of phosphocreatine (PCr) with ADP to produce creatine (Cr) and ATP, and (d) adenylate kinase activity that catalyzes interconversion of adenine nucleotides: 2ADP ⇔ ATP + AMP, and (e) the additional NADH pool produced in cytosol and transported to mitochondria by different shuttle mechanisms. For oxygen consumption, we assumed that mitochondrial respiration (O$_2$ consumption) depends on cellular pyruvate and oxygen concentrations and that respiratory chain activity is activated by ADP concentration.

The state of the system is represented by the state vector of time-dependent metabolite concentrations $C_n$, (n = 1, ..., N) and includes 27 state variables whose names, balance equations, and steady-state values are presented in **Supplementary file 1**. The model also includes three mass conservations laws, that reduce the number of state variables. The temporal profile of the system is governed by the set of ODE based on the model network that distinguishes media and cellular domains. For the metabolite $i$ in the intracellular compartment, the general form is:

$$\frac{dC_i}{dt} = J_i^{tr} + \sum_j v_i^{pj} V_i^{pj} - \sum_k v_i^{uk} V_i^{uk}, \tag{1}$$

where the intracellular concentration of species $i$ is $C_i$; $J_i^{tr}$ is the net transport flux for boundary species $i$ between the media and cell; $V_i^{pj}$, $V_i^{uk}$ are normalized reaction fluxes that produce ($j$) or utilize ($k$) cellular species $i$, $v_i^{pj}$, and $v_i^{uk}$ are corresponding stoichiometric coefficients. For extracellular boundary metabolites in the media, the dynamic mass balance has the following general form:

$$\frac{dCe}{dt} = r_{ie} \cdot J_i^{tr}, \tag{2}$$

where the concentration of extracellular ($e$) species is $C_e$ and $r_{ie}$ is the ratio of cell volume to media volume. To ensure that steady states are obtained during perturbations of parameters without loss of generality, we assume that the media boundary metabolites have constant concentrations. The transport of boundary species is taken to be either facilitated (glucose, lactate, serine, glycine) or passive (oxygen). For passive diffusion of oxygen the equation for the net transport flux is:

$$J_{O_2}^{tr} = k_{O_2} \left( C_{O2e} - C_{O2i} \right), \tag{3}$$

where $k_{O_2}$ is the effective rate constant for passive $O_2$ diffusion, and $C_{eO2}$ and $C_{iO2}$ are medium and intracellular oxygen concentrations. For facilitated transport, the equation for the net transport flux is:

$$J_i^{tr} = \frac{J_{tr}^{max} \left( \dfrac{C_{ei}}{K_{mi}^{tr}} - \dfrac{C_i}{K_{mi}^{tr}} \right)}{1 + \dfrac{C_{ei}}{K_{mi}^{tr}} + \dfrac{C_i}{K_{mi}^{tr}}}, \tag{4}$$

where $C_{ei}$ is the extracellular concentration for species $i$, $J_{tr}^{max}$ is the maximal transport rate, and $K_{mi}^{tr}$ is the Michaelis–Menten constant for transport. NADH transport from the cytosol to the mitochondria is

mediated by the malate–aspartate and glycerol phosphate shuttles. We modeled the NAD⁺-mediated mitochondrial shuttle flux from NADH to NAD⁺ by assuming that the total shuttle flux is balanced with the NADH-generation reactions involving LDH and PHGDH.

$$V_{mas} = V_{ox} + 2V_{phgdh} \qquad (5)$$

When possible, in order to minimize the number of parameters, we utilized a 'one-step binding enzyme mechanism' with a rate law of the form (**Segel, 1975**):

$$V = \left( \frac{V_f^{max} \dfrac{\prod_i C_{si}^{v_i}}{Kmf} - V_r^{max} \dfrac{\prod_j C_{pj}^{v_j}}{Kmr}}{1 + \dfrac{\prod_i C_{si}^{v_i}}{Kmf} + \dfrac{\prod_j C_{pj}^{v_j}}{Kmr}} \right) \chi(a, \imath), \qquad (6)$$

where $\chi(a, \imath)$ is a control function that accounts for the effects of activation $a$, or inhibition $\imath$. A general form for the control function is (**Liebermeister and Klipp, 2006**):

$$\chi(a, \imath) = \begin{cases} [A] \Big/ ([A] + K_A) & \text{in case of activation,} a \\ \\ K_I \Big/ ([I] + K_I) & \text{in case of inhibition,} \imath \end{cases}, \qquad (7)$$

where [A] and [I] are the concentrations of the activator or inhibitor, $K_A$ and $K_I$ are the corresponding activation and inhibition constants. With this convention, there are 4 (or 5 including the control function) independent parameters for each metabolic flux with a one-step binding mechanism. For fluxes with no allosteric regulation, $\chi(a, \imath) = 1$.

For all metabolic fluxes involving NAD⁺/NADH oxidoreductase activity, we utilized a more complex rate law than that of a one step binding reaction (**Cornish-Bowden, 2012**). We chose to consider a more complicated rate-law since our initial observations indicated that a one-step binding mechanism produced numerical instabilities in the solutions. We therefore considered a general rate law corresponding to a random bi–bi mechanism (for LDH and PHGDH) and a random ter-bi reaction for GAPDH. For example, in its general form the rate law for LDH is:

$$V_{ldh} = \frac{V_{fldh}^{max} \dfrac{PYR * NADH}{Kmldhf} - V_{rldh}^{max} \dfrac{LAC_i * NAD^+}{Kmldhr}}{1 + \dfrac{PYR}{K_{mpyr}^{ldh}} + \dfrac{NADH}{K_{mnadh}^{ldh}} + \dfrac{PYR * NADH}{Kmldhf} + \dfrac{LAC_i}{K_{mlac}^{ldh}} + \dfrac{NAD^+}{K_{mnad}^{ldh}} + \dfrac{LAC_i * NAD^+}{Kmldhr}}. \qquad (8)$$

However, it has also been suggested that the enzymatic reaction for oxidoreductase activity proceeds in an ordered fashion resulting in a rate law of the form:

$$V_{ldh} = \frac{V_{fldh}^{max} \dfrac{PYR * NADH}{Kmldhf} - V_{rldh}^{max} \dfrac{LAC_i * NAD^+}{Kmldhr}}{1 + \dfrac{NADH}{K_{mnadh}^{ldh}} + \dfrac{PYR * NADH}{Kmldhf} + \dfrac{NAD^+}{K_{mnad}^{ldh}} + \dfrac{LAC_i * NAD^+}{Kmldhr}} \qquad (9)$$

When considering this mechanism, it was found that statistical evaluation of the model was robust to changes in the choice of rate law.

The rate equations for each metabolic reaction and metabolic parameters values are in **Supplementary file 1**. Taking into account additional thermodynamic constraints by invoking the Haldane equation that relates the forward and reverse fluxes and their Vmax values allowed for a further reduction in the number of independent parameters to three parameters per reaction flux. The Haldane equation has the form:

$$V_r^{max} = \frac{V_f^{max} Kmr}{Kmf K_{eq}}, \qquad (10)$$

where $K_{eq}$ is the reaction apparent equilibrium constant: $K_{eq} = exp(-\Delta G^\circ/RT)$, where $\Delta G^\circ$ is standard reaction Gibbs free energy. $\Delta G^\circ$ values for glycolytic reactions are listed in **Supplementary file 1**.

We first used two independent sets of experimental data from a compendium of steady-state metabolite concentrations of liver cells and steady-state fluxes of DB1 melanoma cells evaluated using a cultured bioreactor (**Konig et al., 2012**; **Shestov et al., 2013**). The model was further validated by a comparison of measurements of redox status, assessment of ATP, ADP, and AMP concentrations. Together, those data were able to generate a thermodynamically feasible and experimentally observable model of glycolysis that was then used as the starting point for exhaustive Monte Carlo simulations and coupled metabolic control analysis.

## Monte Carlo simulations for global sensitivity analysis

The specific aim of this work is to enumerate the variables within and coupled to glycolysis that determine the extent of aerobic glycolysis. We therefore developed a novel algorithm to assess these features. The algorithm involves a global sensitivity analysis based on a Monte Carlo method. The Monte Carlo analysis allows for sampling of parameter space over a broad range of simultaneous variations of parameters (enzyme expression levels) followed by statistical assessment of the resulting solution. The idea of this method is to inject uncertainty of the parameters in the model by randomly selecting parameter values from uniform probability distributions. This was achieved by a Monte Carlo method using a randomly drawn set of parameters. The range of parameters chosen with uniform sampling was within two orders of magnitudes of each Vmax (which is proportional to enzyme activity level and was in the range of 10 times less and 10 times greater than the initial Vmax value). Such a sampling was chosen to be large enough to cover all feasible solutions of aerobic glycolysis. The sampling was carried out using a Latin Hypercube sampling method (**Oguz et al., 2013**). For each chosen set of random parameter vectors the model was simulated between 2000 and 5000 times and its output was calculated. Briefly, we divided the range of the *i-th* normalized parameter into n (n = 2000–5000) subintervals of equal size. Then, we randomly sample n values (e.g., $p_i$, i = 1, …, n), one from each subinterval, for the *i-th* parameter. We next randomly permuted the n values for each parameter to get the parameter vector. We then evaluated the following: the Warburg Effect value, energy and redox states, metabolite steady-state concentrations and fluxes and flux control coefficients (FCCs). The flux control coefficient that an enzyme $E_i$ exerts on the lactate flux $J_{Lac}$ is defined as (**Heinrich and Rapoport, 1974**; **Fell, 1992**; **Shestov et al., 2013**):

$$FCC_i = \frac{\partial \ln J_{Lac}}{\partial \ln E_i}. \tag{11}$$

To compute the FCC values numerically, we considered a perturbation 0.01 times of the enzyme value and then evaluated the change in flux. The model was implemented in Matlab as a system of 27 ordinary differential equations (ODEs) with an ODE solver designed for stiff ODE systems.

## Cell culture and metabolite extraction

HCT116 cells were cultured with a growth medium, which contains RPMI 1640, 10% Fetal Bovine Serum (FBS), 100 U/ml penicillin, and 100 µg/ml streptomycin. The cells were obtained as a gift from Lewis Cantley's laboratory. The cells were cultured in 37°C with 5% $CO_2$. For treatments, the cells were seeded in 6-well plate at a density of $2 \times 10^5$ to $5 \times 10^5$ cells per well. After overnight incubation, full growth media were removed, and cells were washed with 2 ml PBS before the addition of RPMI media (without glucose), supplemented with 10% dialyzed and heat inactivated FBS, 100 U/ml penicillin, 100 µg/ml streptomycin and 5 mM 13C-U-glucose (Cambridge Isotope Laboratory, Tewksbury, MA) (fresh growth media). For drug treatments and flux measurements, full growth media were replaced with fresh growth media, containing either 0.1% DMSO, or the concentration of the indicated drug. Media were then collected at 30, 60, 90, and 120 min after incubation. From 20 µl media, 80 µl of ice cold $H_2O$ was added, together with 400 µl ice cold methanol (Fisher, Optima LC/MS grade). After vigorous vortexing, the solution was then centrifuged at 20,000×g at 4°C for 10 min and then the supernatant was dried under vacuum. At 120 min, the media were removed as completely as possible, and then 6-well plates were immediately placed on dry ice, followed by the addition of 1 ml extraction solvent, 80% MeOH/$H_2O$ (Fisher, Optima LC/MS Grade), which was pre-cooled in −80°C freezer for at least 1 hr. The dishes were then transferred to the −80°C freezer. The plates were left for 15 min and then cells were scraped into the solvent

on dry ice and then transferred to two 1.7-ml eppendorf tubes, and centrifuged with the speed of 20,000×*g* at 4°C for 10 min. Metabolite extracts are prepared from three separate wells to make three replicate samples. The supernatant is then transferred to new eppendorf tubes, and dried under vacuum. For analysis, each extract was then re-constituted into water (15 µl for cell extract and 50 µl for medium extract) and then diluted with an equal volume of 50% Methanol/Acetonitrile. Finally, 5 µl was injected into the column for analysis.

## Flux analysis

Absolute concentrations of $^{13}$C lactate were first measured by mixing a standard of unlabeled lactate with a defined concentration into the medium. For each calculated flux, an estimation of the flux from glucose to lactate was obtained from the slope of the time course of $^{13}$C lactate production using the time points of media collection described above. Slopes were computed using Graphpad Prism with time courses determined to be linear from the goodness of fit ($R^2 > 0.98$ with few exceptions). For corresponding plots for different agents, fraction of inhibition was computed from $K_i$ values that were taken based on previous reports (*Foxall et al., 1984*; *Le et al., 2010*; *Telang et al., 2012*).

## Mass spectrometry

The Q Exactive Mass Spectrometer (QE-MS) is equipped with a heated electrospray ionization probe (HESI), and the relevant parameters are as listed: heater temperature, 120°C; sheath gas, 30; auxiliary gas, 10; sweep gas, 3; spray voltage, 3.6 kV for positive mode, and 2.5 kV for negative mode. Capillary temperature was set at 320°C, and S-lens was 55. A full scan range from 60 to 900 (*m/z*) was used. The resolution was set at 70,000. The maximum injection time was 200 ms with typical injection times around 50 ms. These settings resulted in a duty cycle of around 550 ms to carry out scans in both positive and negative mode. Automated gain control (AGC) was targeted at 3,000,000 ions.

## Liquid chromatography

Liquid chromatography (Ultimate 3000 UHPLC) is coupled to the QE-MS for metabolite separation and detection. An Xbridge amide column (100 × 2.1 mm i.d., 3.5 µm; Waters) is employed for compound separation. The mobile phase A is 20 mM ammonium acetate and 15 mM ammonium hydroxide in water with 3% acetonitrile, pH 9.0, as described above, and mobile phase B is acetonitrile. A linear gradient was used as follows: 0 min, 85% B; 1.5 min, 85% B; 5.5 min, 35% B; 10 min, 35% B; 10.5 min, 35% B; 14.5 min, 35% B; 15 min, 85% B; and 20 min, 85% B. The flow rate was 0.15 ml/min from 0 to 10 min and 15 to 20 min, and 0.3 ml/min from 10.5 to 14.5 min. All solvents are LC-MS Optima grade and purchased from Fisher Scientific.

## NADH/NAD$^+$ imaging

Human mammary epithelial MCF-10A cells (CRL-10317; ATCC) stably expressing Peredox-NLS were generated and cultured as previously described (*Debnath et al., 2003*; *Hung et al., 2011*). 2–4 days prior to imaging, ~500 cells were plated onto the center of each well of a 96-well plate. On the day of the experiment, cells were placed in custom DMEM/F12 (Gibco) containing no glucose and supplemented to the levels mentioned. Fluorescence images were acquired using a Nikon inverted Eclipse Ti microscope, equipped with a Nikon 20×/0.75 Plan Apo objective, three different regions of interest chosen, images sequentially acquired every 8 min with 50–100 ms exposure and 2 × 2 binning. Using a custom MATLAB algorithm previously developed, we subtracted background, set a threshold for cell segmentation, and analyzed the data as previously described (*Hung et al., 2011*). The concentration of glucose is maintained by considering a culture system in which cells are seeded at low density and media are present in vast excess. An estimate of glucose maintenance in the media in the low glucose condition can be considered using values the glucose uptake rate of the cells (~100 fmol/cell/hr), the cell number (~2000), the volume of media used (400 µl), and concentration of glucose in the media (750 µM, 500 µM from supplementation + 250 µM from the Horse Serum). Thus in 24 hr, we estimate that about 5 nmols of glucose are consumed. The media contain roughly 200 nmols of glucose which is in excess of the amount consumed by the cells.

## Data analysis

Raw data collected from the QE-MS is processed using Sieve 2.0 (Thermo Scientific). Peak alignment and detection are performed according to the protocol described by Thermo Scientific. For a targeted

metabolite analysis, the method 'peak alignment and frame extraction' is applied. An input file of theoretical *m/z* and detected retention time known metabolites is used for targeted metabolite analysis with data collected in both positive and negative mode. *m/z* width is set at 10 ppm. The output file including detected *m/z* and relative intensity in different samples is obtained after data processing. For resulting simulation data, hierarchical clustering was carried out using spearman ranked correlations and the Gene-e software package (Broad Institute). Box-plots (25%/75% percentile, mean and median) was calculated and made with the Graphpad Prism software package.

## Acknowledgements
We are grateful to Chi V Dang, Navdeep Chandel, Dimitris Anastasiou, and members of the Locasale lab for helpful comments.

## Additional information

### Competing interests
JWL: A patent related to this work has been filed. US Provisional Patent Appln. No. 61/908,953. The other authors declare that no competing interests exist.

### Funding

| Funder | Grant reference number | Author |
| --- | --- | --- |
| National Institutes of Health | R00CA168997 | Jason W Locasale |
| National Institutes of Health | R01AI110613 | Jason W Locasale |
| International Life Sciences Institute Research Foundation | Future Leader Award | Jason W Locasale |

The funders had no role in study design, data collection and interpretation, or the decision to submit the work for publication.

### Author contributions
AAS, XL, Conception and design, Acquisition of data, Analysis and interpretation of data; ZS, AAC, YPH, LH, DK, JGA, Acquisition of data, Analysis and interpretation of data; AL, Contributed to the analysis of the experiments with FX11, Contributed unpublished essential data or reagents; GY, Contributed to the NADH/NAD+ measurements, Contributed unpublished essential data or reagents; JWL, Conception and design, Acquisition of data, Analysis and interpretation of data, Drafting or revising the article

### Author ORCIDs
Jason W Locasale, http://orcid.org/0000-0002-7766-3502

## Additional files

### Supplementary file
• Supplementary file 1. Equations and parameters used for the glycolysis model.

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
