## [Decision Letter]

Thank you for sending your work entitled “A quantitative model of aerobic glycolysis identifies GAPDH as a rate-limiting step” for consideration at *eLife.* Your article has been favorably evaluated by Charles Sawyers (Senior editor), 2 experts in the field of cancer metabolism, as well as a member of our Board of Reviewing Editors.

The Reviewing editor and the other reviewers discussed their comments before we reached this decision, and the Reviewing editor has assembled the following comments to help you prepare a revised submission.

All the reviewers and editors agree that the work is interesting as it addresses how glycolytic rate is controlled, a question lacking conclusive answers. The experiments are rigorous and the results are for the most part convincing. However, there are a number of concerns that need to be addressed before the manuscript can be accepted.

1) An important conclusion is that GAPDH exerts a surprising level of control over glycolytic rate and thus it would be important to expand upon how GAPDH constrains flux. Is the bottleneck imposed simply by reduced levels of enzyme expression, in which case over-expressing the enzyme should produce major changes in metabolite levels and overall flux? Or are other effects at work, such as ROS-mediated inhibition, which has been observed in several systems?

2) A concern is that the specificity of the glycolytic inhibitors used is questionable. This is a general issue for the field and not easily addressed by these authors, but should be acknowledged in interpreting the experiments.

3) Measuring glucose consumption directly would be a more straightforward way to assess changes in overall glycolytic flux, and addresses aspect of the Warburg effect (increased glucose uptake) that is not considered by this work.

4) Distributions of NADH/NAD+ ratios are presented for cells at different glucose concentrations, but how glucose concentrations were maintained in culture over a 24-hour experiment is not clear. Particularly for the 0.5 mM glucose sample, was the concentration of glucose still 0.5 mM when ratios were measured? This seems unlikely.

5) A straightforward explanation for why GAPDH activity predicts lactate production is that these steps are linked by NAD+ use and regeneration. Even in the absence of modeling a consideration of glycolysis suggests this relationship yet it is hardly mentioned. This point needs to be discussed.

---

## [Author Response]

1) An important conclusion is that GAPDH exerts a surprising level of control over glycolytic rate and thus it would be important to expand upon how GAPDH constrains flux. Is the bottleneck imposed simply by reduced levels of enzyme expression, in which case over-expressing the enzyme should produce major changes in metabolite levels and overall flux? Or are other effects at work, such as ROS-mediated inhibition, which has been observed in several systems?

The reviewers raise important points that warrant further discussion. We first note that many different mechanisms can lead to changes in GAPDH activity. For example, one instance was investigated in Figure 4 in which flux control over glycolysis was measured by direct inhibition by a small molecule of GAPDH in which case it was shown to cause the largest decrease in glycolytic flux as opposed to inhibition at other steps of the pathway.

Furthermore, increases in GAPDH expression are also predicted to increase flux through the pathway although as the model predicts the correlation is small (R2 ∼0.4) so this will not always be the case. In fact, GAPDH is the most abundant enzyme in glycolysis suggesting that in many cases the expression of glycolytic enzymes is designed to achieve optimal flux through the pathway (see Moghaddas, Cell Reports 2013, now cited in the text). An increase in activity can occur through multiple mechanisms including the alleviation of its inhibition by ROS or other mechanisms, increasing its activity through allosteric activation, or by increasing its expression which proportionately increases the enzyme activity through increasing Vmax. Together each of these possibilities would manifest themselves in different contexts. We have included the mention of these multiple possibilities in the Discussion. A thorough discussion of each of these effects is mentioned in a revised manuscript.

*2) A concern is that the specificity of the glycolytic inhibitors used is questionable. This is a general issue for the field and not easily addressed by these authors, but should be acknowledged in interpreting the experiments*.

We thank the reviewers for raising this important point. We have modified the text to include caveats regarding the specificity of the inhibitors, resulting limitations in the interpretation of the data, and our efforts to minimize these confounding variables.

*3) Measuring glucose consumption directly would be a more straightforward way to assess changes in overall glycolytic flux, and addresses aspect of the Warburg effect (increased glucose uptake) that is not considered by this work*.

We thank the reviewers for this suggestion. In designing this study, we reasoned that a more conclusive measurement of glycolytic flux was the rate at which glucose was being converted to lactate. This measurement accounts for both glucose uptake and production of lactate. We reasoned that glucose uptake alone does not account for the portion of glycolytic flux that is siphoned into biosynthetic pathways or into the mitochondria. Nevertheless, we have also measured additional fluxes (Author response image 1) including glucose uptake rate, and a biosynthesis rate from glucose and found that each does correlate reasonably well with the rate of glucose to lactate. This result is shown below:Author response image 1.Comparison of glucose uptake rate and rate from glucose to lactate production. A) Diagram of flux experiment. Flux from glucose to pyruvate (e.g. glucose uptake), serine (biosynthetic flux), and lacate (Warburg Effect) is measured. ^13^C glucose is input as described in the methods. B) Comparison of fluxes for 3PO (PFK2 inhibition) treatment C) Comparison of fluxes for IA (GAPDH inhibition) treatment. D) Comparison of fluxes for FX11 (LDHA inhibition) treatment. E) Correlation between flux to serine and flux to lactate across each measurement. F) Correlation between flux to pyruvate and flux to lactate across each measurement.

*4) Distributions of NADH/NAD+ ratios are presented for cells at different glucose concentrations, but how glucose concentrations were maintained in culture over a 24-hour experiment is not clear. Particularly for the 0.5 mM glucose sample, was the concentration of glucose still 0.5 mM when ratios were measured? This seems unlikely*.

We appreciate this concern and have included further clarifications in the Methods. By our calculations shown below and now contained in the Methods, the glucose concentration should be close to constant over the duration of the experiment. The concentration of glucose is maintained by considering a culture system in which cells are seeded at low density and media is present in vast excess. A quantitative estimate of glucose maintenance in the media can be considered using values of the glucose uptake rate of the cells (∼100fmol/cell/hr), the cell number (∼2000), the volume of media used (400 μL), and concentration of glucose in the media (500μM). Thus in 24 hours, we estimate that about 5nmols of glucose are consumed. The media contains about 200nmols of glucose which is in excess of the amount consumed by the cells.

*5) A straightforward explanation for why GAPDH activity predicts lactate production is that these steps are linked by NAD+ use and regeneration. Even in the absence of modeling a consideration of glycolysis suggests this relationship yet it is hardly mentioned. This point needs to be discussed*.

In the Discussion, we have expanded on the mechanisms of GAPDH exerting control over glycolytic rate. The mechanism is partially explained by NAD+ use and regeneration but the full explanation appears more complicated and is due to GAPDH existing at a unique point in the pathway where it can be affected by redox (NAD+/NADH), energy (ATP consumption, and carbon flux (BPG and G3P) that together allow for more points of regulation.

However we respectfully disagree with the reviewers that this finding could be obtained without modeling especially since it was not previously known that GAPDH exerts control over glycolysis. In fact numerous introductory biochemistry textbooks for undergraduate and medical school students have discussed in detail the control of glycolysis and the emphasis has been in these discussions on the steps in glycolysis that involve changes in standard free energies such as pyruvate kinase or phosphofructokinase.